# Structural Characterization of Full-Length Human Dehydrodolichyl Diphosphate Synthase Using an Integrative Computational and Experimental Approach

**DOI:** 10.3390/biom9110660

**Published:** 2019-10-28

**Authors:** Michal Lisnyansky Bar-El, Su Youn Lee, Ah Young Ki, Noa Kapelushnik, Anat Loewenstein, Ka Young Chung, Dina Schneidman-Duhovny, Moshe Giladi, Hadas Newman, Yoni Haitin

**Affiliations:** 1Department of Physiology and Pharmacology, Sackler Faculty of Medicine, Tel-Aviv University, Tel Aviv 6997801, Israel; michallis505@gmail.com; 2School of Pharmacy, Sungkyunkwan University, Jangan-gu, Suwon 16419, Korea; youn3887@hanmail.net (S.Y.L.); kay9039@naver.com (A.Y.K.); kychung2@gmail.com (K.Y.C.); 3Department of Ophthalmology, Sheba Medical Center, Ramat Gan 5265601, Israel; kapelushniknoa@gmail.com; 4Department of Ophthalmology, Tel-Aviv Sourasky Medical Center, Tel Aviv 6423906, Israel; anatl@tlvmc.gov.il (A.L.); hadasng@gmail.com (H.N.); 5Department of Biological Chemistry, Institute of Life Sciences, The Hebrew University of Jerusalem, Jerusalem 9190401, Israel; dina.schneidman@mail.huji.ac.il; 6School of Computer Science and Engineering, The Hebrew University of Jerusalem, Jerusalem 9190401, Israel; 7Tel Aviv Sourasky Medical Center, Tel Aviv 6423906, Israel

**Keywords:** *cis*-prenyltransferase, computational modeling, hydrogen–deuterium exchange mass-spectrometry, small-angle X-ray scattering, enzyme kinetics, DHDDS

## Abstract

Dehydrodolichyl diphosphate synthase (DHDDS) is the catalytic subunit of the heteromeric human *cis*-prenyltransferase complex, synthesizing the glycosyl carrier precursor for N-linked protein glycosylation. Consistent with the important role of N-glycosylation in protein biogenesis, DHDDS mutations result in human diseases. Importantly, DHDDS encompasses a C-terminal region, which does not converge with any known conserved domains. Therefore, despite the clinical importance of DHDDS, our understating of its structure–function relations remains poor. Here, we provide a structural model for the full-length human DHDDS using a multidisciplinary experimental and computational approach. Size-exclusion chromatography multi-angle light scattering revealed that DHDDS forms a monodisperse homodimer in solution. Enzyme kinetics assays revealed that it exhibits catalytic activity, although reduced compared to that reported for the intact heteromeric complex. Our model suggests that the DHDDS C-terminus forms a helix–turn–helix motif, tightly packed against the core catalytic domain. This model is consistent with small-angle X-ray scattering data, indicating that the full-length DHDDS maintains a similar conformation in solution. Moreover, hydrogen–deuterium exchange mass-spectrometry experiments show time-dependent deuterium uptake in the C-terminal domain, consistent with its overall folded state. Finally, we provide a model for the DHDDS–NgBR heterodimer, offering a structural framework for future structural and functional studies of the complex.

## 1. Introduction

Protein N-glycosylation is a critical post-translational modification, in which an oligosaccharide moiety is transferred from the glycosyl carrier dolichol-phosphate (Dol-P) and covalently attaches to a polypeptide chain via an asparagine residue [1]. Following protein conjugation, the oligosaccharide is further modified, increasing the functional and structural diversity of the proteome [2]. Indeed, this modification is crucial for proper protein folding, oligomerization, quality control, sorting, and transport [2]. Accordingly, even slight variations in protein glycosylation can markedly affect proteins structure and function [3] and, thus, glycosylation disorders result in a wide range of clinical syndromes affecting practically every organ system [4,5].

Dehydrodolichyl diphosphate synthase (DHDDS) together with Nogo-B receptor (NgBR) form the human *cis*-prenyltransferase complex [6,7]. This complex catalyzes the formation of dehydrodolichyl diphosphate (DHDD), a long-chain polyprenyl serving as a precursor for Dol-P. DHDD is synthesized by chain elongation of farnesyl diphosphate (FPP) via multiple condensations with isopentenyl diphosphate (IPP) (Figure 1A) [6]. Importantly, while previous studies revealed that DHDDS is required and sufficient for this catalytic activity, NgBR induces an increase in the expression and activity of the complex but exhibits no catalytic activity [6,7,8]. In line with the important role of DHDDS in protein glycosylation, mutations in DHDDS were recently shown to cause autosomal recessive retinitis pigmentosa [9,10], a fatal glycosylation disorder [11], and developmental epileptic encephalopathies [12].

Despite the clinical importance of DHDDS, it was not structurally and functionally characterized to date. Conserved domains search analysis [13] reveals that the *cis*-prenyltransferase catalytic domain of DHDDS (residues 24–248) is homologous to undecaprenyl pyrophosphate synthase (UPPS), a well characterized bacterial *cis*-prenyltransferase [14]. Consisting of seven α-helices and six β-strands, this domain forms dimers, with a hydrophobic catalytic tunnel surrounded by two α-helices and four β−strands within each monomer [15]. In contrast with the N-terminal *cis*-prenyltransferase homology domain, the C-terminus of human DHDDS does not converge with any known conserved domains. Intriguingly, despite the low level of conservation, a C-terminal domain is common to long-chain *cis*-prenyltransferases and is absent in short- and medium-chain *cis*-prenyltransferases [7] (Figure 1B). Nevertheless, its structure and functional role remains elusive.

Recently, we have devised an efficient procedure for overexpression and purification of human DHDDS in *Escherichia coli* [16,17]. Similar to the bacterial homolog UPPS, we showed that human DHDDS forms active homodimers [17]. However, the molecular details of the interaction interface and the overall architecture of the enzyme remained obscure. Here, we determine a structural model for the full-length human DHDDS homodimer using an integrative approach that combines multiple sources of information [18], including a radioligand-based enzyme kinetics assay, size-exclusion chromatography multi-angle light scattering (SEC-MALS), small-angle X-ray scattering (SAXS), and hydrogen–deuterium exchange mass-spectrometry (HDX-MS).

## 2. Materials and Methods

### 2.1. Cloning

Full-length human DHDDS (UniProt Q86SQ9) was synthesized and cloned into pET-32b plasmid between the NdeI and BamHI restriction sites (GenScript, USA) as a thioredoxin (TRX) fusion protein, as previously described [17]. The construct includes a 6xHis-tag to facilitate protein purification and a TEV-protease (tobacco etch virus) cleavage site to remove the 6xHis-tag and TRX fusion.

### 2.2. Protein Expression and Purification

Protein expression and purification was carried out as previously described [16,17]. Briefly, *E. coli* T7 express competent cells were transformed, grown in 2xYT medium at 37 °C until reaching OD_595nm_ = 0.5 and induced at 16 °C by adding 0.5 mM isopropyl β-D-1-thiogalactopyranoside (IPTG). Proteins were expressed at 16 °C for 16–20 h, harvested by centrifugation (10,000× *g* for 10 min), and then resuspended in buffer with 1 μg/mL DNase I (type DN-25, Sigma) and a protease inhibitor mixture (Roche Applied Science, Germany). Resuspended cells were homogenized and disrupted in a microfluidizer (Microfluidics, USA). Soluble proteins were recovered by centrifugation at ~40,000× *g* for 45 min at 4 °C. Overexpressed proteins were purified on a TALON-Superflow column (GE healthcare, UK), followed by TEV protease cleavage of the 6xHis-tag. The reaction mixture was buffer exchanged to remove imidazole and loaded again on a TALON-Superflow column. The unbound protein was collected and loaded onto a Superdex-200 preparative size-exclusion column pre-equilibrated with Tris-HCl, pH 7.5, 150 mM NaCl, 20 mM β-mercaptoethanol, and 0.02% Triton X-100. Purified proteins were flash-frozen in liquid nitrogen and stored at −80 °C until use. Protein purity was >90%, as judged by SDS-PAGE.

### 2.3. SEC-MALS

Experiments were carried out using a size-exclusion chromatography column (superdex-200 increase 10/300 GL column) pre-equilibrated with Tris-HCl, pH 7.5, 150 mM NaCl, 20 mM β-mercaptoethanol, and 0.02% Triton X-100. Samples (2 mg/mL; 50 µL) were injected onto an HPLC, connected to an eight-angle light-scattering detector, followed by a differential refractive-index (RI) detector (Wyatt Technology, Santa Barbara, CA, USA). RI and MALS readings were analyzed with the ASTRA software package (Wyatt Technology, Santa Barbara, CA, USA) to determine molecular mass.

### 2.4. Enzyme Kinetics

The activity of purified DHDDS was assayed as previously described [16,17]. Briefly, 0.1–0.5 μM of purified protein was mixed with FPP and [^14^C]-IPP to initiate the reaction in buffer composed of Tris-HCl, pH 7.5, 150 mM NaCl, 20 mM β-mercaptoethanol, 0.1% Triton X-100 at 30 °C. Then, 10 mM EDTA (final concentration) was added to quench the reaction and 1 mL of water-saturated 1-butanol was added to extract the reaction products by thorough vortexing. Initial rates were measured by quenching the reaction at 10% or lower substrate consumption. The K_m_ value of FPP was determined in the presence of 0.1 μM protein by varying [FPP] while holding [IPP] constant at 20 μM, and the K_m_ value of IPP was determined in the presence of 0.5 μM protein by varying [IPP] while holding [FPP] constant at 10 μM. The product, encompassing ^14^C, was quantitated using a scintillation counter. Kinetic constants were obtained by fitting the data to the Michaelis–Menten equation using Origin 7.0 (OriginLab, USA).

### 2.5. SAXS

SAXS data were measured at beamline BM29 of the European Synchrotron Radiation Facility (ESRF), Grenoble, France. Fifty microliters of purified protein concentrated to ~8 mg/mL was chromatographed using a size-exclusion chromatography column (superdex-200 increase 10/300 GL column) pre-equilibrated with Tris-HCl, pH 7.5, 150 mM NaCl, 20 mM β-mercaptoethanol, and 0.02% Triton X-100 on an ultra-performance liquid chromatograph (Shimadzu Corporation). Data were collected in-line at 20 °C with X-ray beam at wavelength λ = 1.0 Å, and the distance from the sample to detector (PILATUS 1M, Dectris Ltd.) was 2.867 m, covering a scattering vector range (q = 4πsinθ/λ) from 0.0025 to 0.5 Å^−1^, with an exposure time of 1 s per frame. The 2D images were reduced to one-dimensional scattering profiles using the software on site. Frames corresponding to dimeric DHDDS were averaged and subtracted from buffer frames using ScÅtter (http://www.bioisis.net/tutorial/9). The experimental radius of gyration (R_g_) was calculated from data at low q values in the range of qR_g_ <1.3, according to the Guinier approximation: lnI(q) ≈ ln(I(0)) − R_g_^2^q^2^/3 using PRIMUS. The D_max_ value and the Porod volume were derived from the paired-distance distribution function (PDDF or P(r)) calculated using GNOM. The flexibility analysis was performed using ScÅtter [19].

### 2.6. HDX-MS

The purified proteins were prepared at a concentration of 50 pmole/µL (50 µM) in a buffer composed of 25 mM Tris/HCl, pH 7.4, 150 mM NaCl, 10 mM β-mercaptoethanol, and 0.02% Triton X-100. The hydrogen–deuterium exchange was initiated by mixing 3 µL of purified protein with 27 µL of the D_2_O buffer (25 mM Tris/HCl, pH 7.4, 150 mM NaCl, 10 mM β-mercaptoethanol, 0.02% Triton X-100, and 10% glycerol in D_2_O) containing 2 mM EDTA; the mixtures were incubated at various time intervals including 10, 100, 1000, and 10,000 s at 4 °C. At the indicated time-points, the mixtures were quenched by adding 30 µL of ice-cold quench buffer (100 mM NaH_2_PO_4_, pH 2.01, 20 mM tris (2-carboxyethyl) phosphine, 1 M guanidine). For non-deuterated samples, 3 µL of the purified protein were mixed with 27 µL of H_2_O buffer (25 mM Tris/HCl, pH 7.4, 150 mM NaCl, 10 mM β-mercaptoethanol, 0.02% Triton X-100, and 10% glycerol in H_2_O) containing 2 mM EDTA, and 30 µL of ice-cold quench buffer was added. The quenched samples were digested online by passing them through an immobilized pepsin column (2.1 mm × 30 mm) (Life Technologies) at a flow rate of 100 µL/min with 0.05% formic acid in water at 12 °C. Peptide fragments were subsequently collected on a C_18_ VanGuard trap column (1.7 µm × 30 mm) (Waters) for desalting with 0.05% formic acid in water and then separated by ultra-performance liquid chromatography (UPLC) using an ACQUITY UPLC C_18_ column (1.7 µm, 1.0 mm × 100 mm) (Waters) at a flowrate of 40 µL/min with an acetonitrile gradient by using two pumps, which started with 8% B and increased to 85% B over the next 8.5 min. The mobile phase A was 0.15% formic acid in water and solvent B was acetonitrile containing 0.15% formic acid. To minimize the back-exchange of deuterium to hydrogen, the sample, solvents, trap, and UPLC column were all maintained at pH 2.5 and 0.5 °C during analysis. Mass spectral analyses were performed with a Xevo G2 Quadruple-Time of Flight (Q-TOF) equipped with a standard electrospray ionization (ESI) source in MS^E^ mode (Waters, Milford, MA, USA). The mass spectra were acquired in the range of m/z 100–2000 for 12 min in positive ion mode. Peptides were identified in non-deuterated samples with ProteinLynx Global Server (PLGS) 2.0 (Waters, Milford, MA, USA). The following parameters were applied: monoisotopic mass, nonspecific for the enzyme, MS/MS ion searches, automatic fragment mass tolerance, and automatic peptide mass tolerance. Searches were performed with the variable methionine oxidation modification, and the peptides were filtered with a peptide score of 6. To process HDX-MS data, the amount of deuterium in each peptide was determined by measuring the centroid of the isotopic distribution using DynamX 2.0 (Waters, Milford, MA, USA). All data were derived from three independent experiments. The back-exchange level was not corrected because the fully deuterated samples aggregated during the long incubation required.

### 2.7. Structural Modeling

The *cis*-prenyltransferase domain was modeled by comparative modeling [20] using Z,Z-farnesyl diphosphate synthase (PDB 5hxn [21], sequence identity 41%) and undecaprenyl pyrophosphate synthase (PDB 6acs [22], sequence identity 35%) as templates. The C-terminal domain (residues 261–340) was modeled by RaptorX-Contact, which is using co-evolutionary information to predict residue–residue contacts [23]. Residues 18–29 were modeled as α-helix based on the secondary structure prediction. The dimer was assembled in MODELLER v9.18 [20] using known dimerization interface of the *cis*-prenyltransferase domains.

## 3. Results

### 3.1. Human DHDDS Forms a Monodisperse Homodimer

Short- and medium-chain *cis*-prenyltransferases form homodimers via a conserved dimerization interface throughout the phylogenetic tree [7,14]. The structure of these homodimers was extensively studied at high resolution [24]. However, while sharing the conserved *cis*-prenyltransferase homology domain, including the dimerization interface, the stoichiometry of long-chain *cis*-prenyltransferases, such as human DHDDS, was not comprehensively studied [7,25]. Previously, in order to assess the oligomeric state of human DHDDS, we used size-exclusion chromatography (SEC). By comparing with a bovine serum albumin (BSA) standard, we concluded that the elution volume corresponds to that expected for a homodimer [17]. However, relying on the elution volume, which depends on the hydrodynamic radius, provides a mere surrogate of the actual mass. For example, lower than expected elution volume, corresponding to a higher mass estimate, can result from an elongated conformation or high flexibility of the previously uncharacterized C-terminus [26].

Thus, following overexpression and purification of full-length human DHDDS in *E. coli* (Figure 2A), we used SEC-MALS (Figure 2B) [27,28]. Importantly, MALS allows determination of absolute molar mass and average size of molecules based on the intensity of light scattered at different angles, while SEC only serves as a fractionation step in this setup [27,28]. In agreement with our previous studies, the calculated mass is 77.0 ± 4.0 kDa, closely matching the calculated mass of a homodimer (78.3 kDa). Notably, the protein is monodisperse (Mw/Mn = 1.000 ± 0.07) in solution and no other oligomeric species are observed (Figure 2B), making this preparation suitable for downstream structural studies.

### 3.2. Catalytic Activity of Purified DHDDS

Previous studies showed that NgBR-containing heteromeric *cis*-prenyltransferase complexes are formed in cells, and NgBR was suggested to increase the activity of DHDDS within the complex [6]. Interestingly, NgBR encompasses a *cis*-prenyltransferase homology domain, which includes the conserved dimerization interface but lacks residues which are essential for catalysis [8]. Thus, it was suggested that DHDDS and NgBR form heterodimers [8]. However, several combinatorial dimeric complexes may co-exist physiologically—DHDDS homodimers, NgBR homodimers, and DHDDS-NgBR heterodimers. Here, we sought to determine the catalytic activity of DHDDS in the absence of NgBR (Figure 3), to gain insight into the possible mechanism that leads to the increased activity exerted by NgBR. The turnover number (*k*_cat_) of homodimeric DHDDS is 1.1 × 10^−3^ ± 0.1 × 10^−3^ s^−1^, with K_m_ values of 9.3 ± 2.8 and 0.45 ± 0.1 μM for IPP and FPP, respectively (Figure 3A,B). Interestingly, the K_m_ values obtained for the homodimeric DHDDS are similar to those obtained for the heteromeric *cis*-prenyltransferase complex, while the *k*_cat_ value is ~ 500-fold lower in the absence of NgBR [25]. Thus, our results suggest that co-assembly with NgBR does not alter the substrates’ affinity, but allosterically increases the catalytic activity of DHDDS.

### 3.3. Structural Features of DHDDS in Solution

Conserved domains search of human DHDDS readily identified the canonical *cis*-prenyltransferase domain but yielded no hits for the C-terminus [13]. However, as a C-terminal extension is common to long-chain *cis*-prenyltransferases (Figure 1B), we hypothesized that it may form a structured domain which plays a role in enzyme assembly and function. Thus, in order to gain insight into the spatial organization of the C-terminus, we used SAXS (Table 1). SAXS provides comprehensive structural information about macromolecules in solution including molecular shape, size, and structural variability [19,29]. Moreover, the SAXS profile can be converted into an approximate distribution of pairwise electron distances in the macromolecule (i.e., PDDF) via a Fourier transform.

SAXS data were recorded in-line with SEC, and frames corresponding to homodimeric DHDDS were averaged and background subtracted (Figure 4A). The protein exhibits R_g_ = 31.8 ± 1.4 Å (radius of gyration, the root-mean-square-distance from the center of mass) and D_max_ = 105 ± 11 Å (maximal intramolecular distance). The PDDF indicates that the protein assumes an overall elongated conformation in solution (Figure 4B), and the derived Porod volume matches that expected for a homodimer (Table 1) [30]. Importantly, Kratky plot analysis demonstrates that the protein is well folded, as reflected by the peak observed at low angles and convergence with the X-axis at high angles (Figure 4C) [19,29]. Next, the protein flexibility was assessed using the Porod–Debye law (Figure 4D). The scattering curve shows a clear plateau within the Porod–Debye region with a Porod exponent of 3.6, consistent with an overall compact and rigid particle [19,29]. In summary, our SAXS analysis is consistent with a well-folded and compactly packed C-terminus.

### 3.4. Structural Modeling of Full-Length DHDDS

Following our SAXS analysis, which suggested that the C-terminus forms a structured domain, we sought to determine the structure of full-length DHDDS using a computational modeling approach. While the *cis*-prenyltransferase domain shows high structural conservation, allowing comparative modeling [20], the N-terminus (residues 18–29) was modeled as α-helix based on the secondary structure prediction and the C-terminal domain (residues 254–333) was modeled by co-evolution analysis [23]. Finally, the dimeric assembly was generated using the canonical dimerization interface of the *cis*-prenyltransferase domains (Figure 5A). Importantly, fitting the model against the SAXS data reveals high agreement with χ^2^ = 1.03 (Figure 5B) [31].

### 3.5. HDX-MS Analysis of Human DHDDS

Our SAXS analysis is consistent with our computational model, suggesting that the *cis*-prenyltransferase homology domains dimerize via the canonical dimerization interface. Moreover, it is consistent with tight packing of the C-terminus, which forms a helix–turn–helix fold, against the homology domain (Figure 5). To further experimentally validate our model, and to gain insight into the structural dynamics of DHDDS, we used HDX-MS. In HDX-MS experiments, the protein sample is diluted into a D_2_O-containing buffer. Following dilution, deuterium in the buffer is exchanged with amide hydrogens within the protein peptide backbone [32]. The reaction is quenched at different time points, followed by proteolytic digestion with pepsin and separation of the peptic peptides using reverse-phase chromatography. The eluted peptides are detected using a mass spectrometer, and the degree of deuterium incorporation within each peptide is determined. The exchange rate depends on the conformational dynamics and solvent accessibility of each protein region, where folded and solvent inaccessible regions exhibit low HDX.

The peptic peptides analyzed by a mass spectrometer cover 83.5% of DHDDS (Appendix A). The levels of deuterium uptake at 10, 100, 1000, and 10,000 s are presented on the protein sequence (Figure 6A) and the levels of deuterium uptake at 10 and 10,000 s are color-coded on our model structure (Figure 6B,C). Because we could not correct the back-exchange level due to the aggregation issue of the fully deuterated samples (as discussed in Section 2.6), the highest deuterium uptake was approximately 52% (Appendix A). The average back-exchange level of our HDX-MS system is approximately 30–40%. Therefore, 40–50% deuterium uptake was considered as very high-level deuterium uptake. We carefully examined the deuterium uptake profiles; the residues with uptake plots with high uptake levels (40–50%) even for the short D_2_O incubation (10 sec) were considered to be as highly flexible and/or exposed to the buffer while the residues with uptake plots with low uptake levels (less than 3%) even for the long D_2_O incubation (10,000 s) were considered to be as highly rigid and/or excluded from the buffer; the residues showing gradual increase in deuterium uptake were considered to have relatively dynamic secondary structures and/or relatively less exposed to the buffer.

As expected, the N-terminus (peptide 4–12), the C-terminus (peptide 327–334), and relatively long loop (peptide 203–210) regions showed the highest deuterium uptake (more than 45% at 10 s) as these regions are exposed to the buffer and flexible. The core of the *cis*-prenyltransferase homology domain (peptides 70–79, 71–82, 93–103, 101–112) showed the lowest deuterium uptake (less than 3% at 10,000 s), which suggests that the core of this domain is highly rigid. On the other hand, other regions of the *cis*-prenyltransferase homology domain showed gradual increase of the deuterium uptake suggesting that these regions form secondary structures with relatively dynamic nature.

Consistent with our structural model, the dimerization interface (peptides 170–182, 173–189, 216–226, 220–226) exhibits medium deuterium incorporation with gradual increase (3–25%) although the peptides contain flexible loops, indicating their relative inaccessibility to the bulk solvent. Importantly, highlighting its folded state, the C-terminus (residues 254–333) exhibits an overall time-dependent deuterium incorporation. Thus, as predicted by our computational model, the HDX-MS results support the notion that the monomers interact via the conserved dimerization interface and that the C-terminus forms a folded domain.

## 4. Discussion

Here we provide a structural model for the full-length human DHDDS using a hybrid experimental and computational approach. Following expression and purification (Figure 2A), we showed using SEC-MALS that DHDDS exists as a monodispersed dimer in solution (Figure 2B), making the preparation suitable for downstream experimentation. The purified homodimer is active, exhibiting similar K_m_ values to the values reported for the heteromeric *cis*-PT complex but a markedly reduced *k*_cat_ (Figure 3) [7]. Despite the lack of matching homologous domains, SAXS analysis revealed that the protein is compact and rigid, supporting a notion that the C-terminus forms a folded domain (Figure 4). Co-evolution analysis predicted a helix–turn–helix structure for the C-terminal domain (Figure 5A), and the model was verified by two independent approaches. The model fits the SAXS data very well (Figure 5B), suggesting that the global structure is similar to that existing in solution. Moreover, HDX-MS analysis supports the structural conservation of the dimerization interface and the folded state of the C-terminus (Figure 6).

In contrast to the well-characterized short- and medium-chain *cis*-prenyltransferases, which function as homodimers [15], long-chain *cis*-prenyltransferases and rubber synthases are heteromeric enzymes [7]. The heteromeric human long-chain *cis*-prenyltransferase, composed of DHDDS and NgBR, is gaining increasing biomedical interest due to the identification of mutations in both its components leading to human diseases [9,10,12,33,34]. Rubber synthases are of significant biotechnological and industrial interest, and a recent study showed that, in *Hevea brasiliensis*, the industrial source for latex, the complex is composed of at least three different subunits [35]. Despite the importance of the heteromeric *cis*-prenyltansferases, the molecular mechanisms involved in subunit engagement, as well as the mechanisms determining their product chain length, remain elusive due to lack of structural information [7]. Thus, the goal of the present study was to provide structural insights into the organization of the entire catalytic subunit of the human *cis*-prenyltransferase complex, using an experimentally validated computational approach.

Importantly, our model predicts a novel C-terminal motif, absent from short- and medium-chain homodimeric *cis*-prenyltransferases. However, its structural role in the context of the heteromeric human long-chain *cis*-prenyltransferase complex remains to be determined. According to the structural conservation of the canonical dimerization interface, DHDDS and NgBR may form heterodimers [8]. Therefore, we constructed a homology model of the *cis*-prenyltransferase homology domain of NgBR based on the recently resolved structure of the yeast ortholog, Nus1 [8], and used it to replace one of the DHDDS subunits in our model (Figure 7). This model of the heterodimeric complex suggests that the C-terminal domain is not directly engaged in subunit interactions. This arrangement may suggest that the C-terminal domain can interact with additional proteins, as in the case of rubber synthase. Future studies, focused on the functional role of the C-terminus, are required to understand its role in the context of the heteromeric complex.

Finally, the DHDDS mutations known to result in autosomal recessive retinitis pigmentosa [9,10] and developmental epileptic encephalopathies [12], and NgBR mutations associated with Parkinson’s disease [34] and a congenital glycosylation disorder [33] were mapped on the heterodimer model (Figure 7). Interestingly, all the DHDDS mutations, which are positively charged (K42E, R37H, R211Q), reside in close proximity to the dimeric interface. Moreover, the NgBR mutation resulting in congenital glycosylation disorder, R290H, is also charged and lies in close proximity to the DHDDS disease-causing mutations and catalytic pocket of the enzymatic complex. R290 is part of the conserved C-terminal RxG motif of NgBR [25], which was previously shown to play a critical role in the catalytic activity of the heteromeric human complex. Indeed, the location of R290 on a surface loop may provide it with the structural flexibility needed for interacting with the catalytic pocket and, in turn, to allosterically activate DHDDS. As we show here, the DHDDS homodimers exhibit markedly reduced *k*_cat_ compared to the heteromeric complex [25] (Figure 3). Taken together, the structural proximity of the NgBR C-terminus to the catalytic pocket, its predicted flexibility, and the reduced activity of homodimeric DHDDS compared with the heteromeric complex harboring NgBR, are consistent with the role of NgBR in allosteric activation of DHDDS.

In summary, this study presents the first model of full-length human DHDDS, experimentally validated using two independent approaches. This model sheds light on the previously uncharacterized C-terminus of DHDDS and, together with the homology model of NgBR, provides a structural framework for future structural and functional studies of the human *cis*-prenyltransferase complex, as well as other long-chain prenyltransferases and rubber synthases.

## Figures and Tables

**Figure 1 biomolecules-09-00660-f001:**
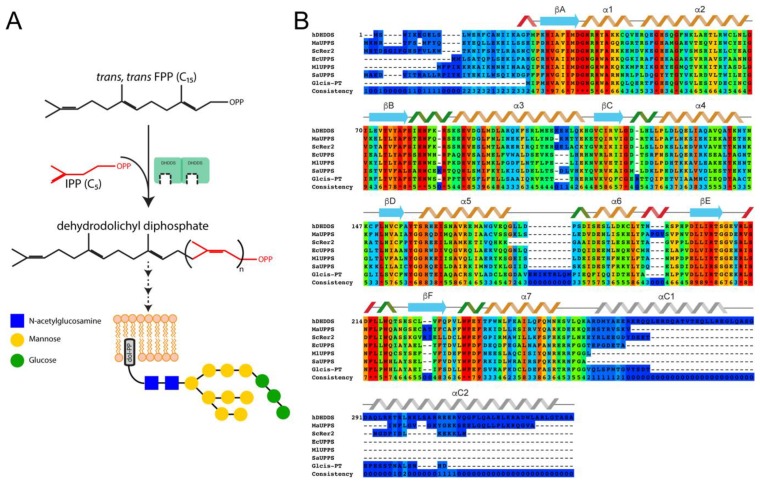
Catalytic cycle of human dehydrodolichyl diphosphate synthase (DHDDS)**.** (**A**) Multiple condensations of a farnesyl diphosphate (FPP) primer with isopentenyl diphosphate (IPP) result in the formation DHDD by DHDDS. DHDD is further processed to form Dol-P, the carrier of the oligosaccharide used for N-glycosylation. (**B**) Sequence alignment of *cis*-prenyltransferases, with conservation scores calculated by PRALINE. Human DHDDS (hDHDDS), *Methanosarcina acetivorans* UPPS (MaUPPS), *Saccharomyces cerevisiae* Rer2 (ScRer2) are components of heteromeric *cis*-prenyltransferase complexes; *Escherichia coli* UPPS (EcUPPS), *Micrococcus luteus* UPPS (MlUPPS), *Sulfolobus acidocaldarius* UPPS (SaUPPS), and *Giardiasis lambia cis*-PT (Glcis-PT) are homodimeric *cis*-prenyltransferases. The secondary structure of the *cis*-prenyltransferase domain is depicted according to the structure of EcUPPS. The secondary structure of the C-terminus is depicted according to our structural prediction.

**Figure 2 biomolecules-09-00660-f002:**
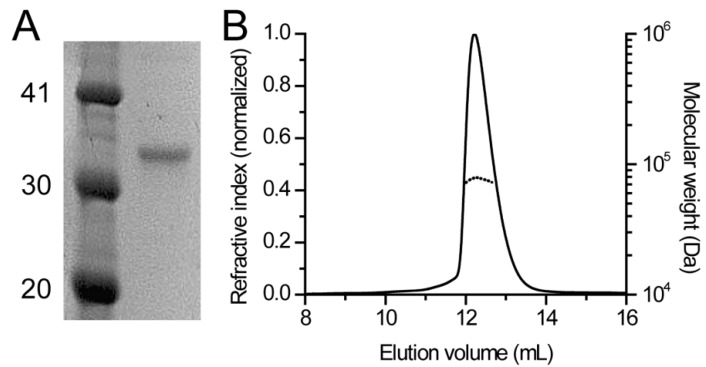
Human DHDDS forms monodisperse homodimers. (**A**) SDS-PAGE analysis of purified DHDDS. Left lane, molecular weight marker; right lane, purified DHDDS. Numbers indicate weight in kDa. (**B**) SEC-MALS analysis of purified DHDDS. The solid line represents the normalized refractive index and the dashed line represents the calculated molecular weight in Da.

**Figure 3 biomolecules-09-00660-f003:**
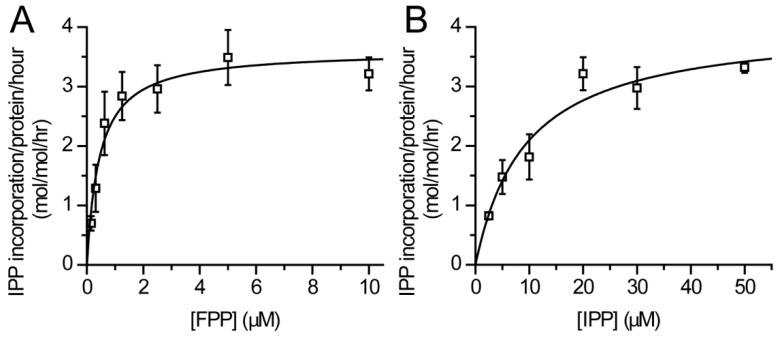
In vitro catalytic activity of purified DHDDS. (**A**,**B**) In vitro activity of purified human DHDDS with FPP and ^14^C-IPP as substrates, assessed as IPP incorporation. (**A**) FPP-dependent activity in the presence of 20 μM ^14^C-IPP. (**B**) IPP-dependent activity in the presence of 10 μM FPP. Initial rates were obtained by quenching the reaction at <10% substrate consumption. Data are presented as mean ± SEM (*n* = 3).

**Figure 4 biomolecules-09-00660-f004:**
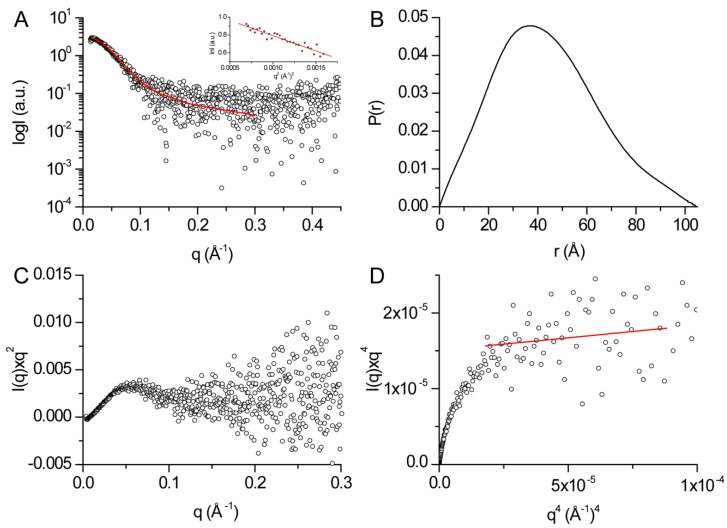
SAXS analysis of purified human DHDDS. (**A**) Experimental SAXS curves of human DHDDS (circles) along with the fit obtained by GNOM (red line). Inset: Guinier analysis. (**B**) Paired-distance distribution function of human DHDDS determined using GNOM. (**C**) Kratky plot analysis. (**D**) Porod–Debye plot. The red line represents the fit used to determine the Porod exponent.

**Figure 5 biomolecules-09-00660-f005:**
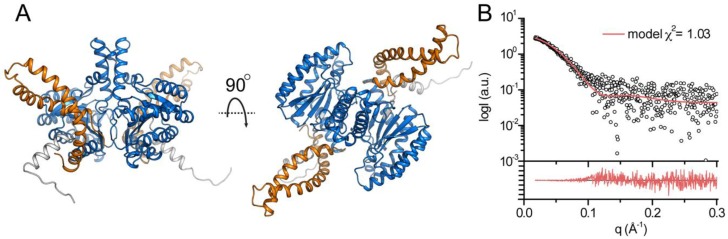
Computational modeling of full-length human DHDDS. (**A**) The model of full-length human DHDDS is shown in cartoon representation. The N-terminus, the *cis*-prenyltransferase homology domain, and the C-terminal helix–turn–helix motif are colored gray, blue, and orange, respectively. (**B**) Fit of the theoretical SAXS curve of the model (red line) to the measured SAXS profile (circles). The lower panel shows the residuals plot.

**Figure 6 biomolecules-09-00660-f006:**
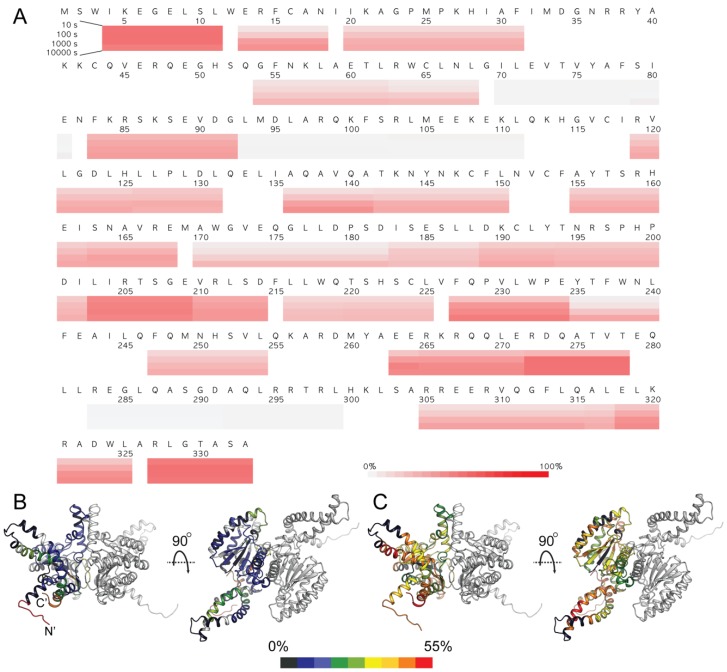
HDX-MS analysis of human DHDDS. (**A**) Deuterium uptake profiles. Incorporation of deuterium at 10, 100, 1000, and 10,000 sec are indicated by color-coded blocks underlining the amino acid sequence. The color legend shows the deuterium uptake levels. (**B**,**C**) The heat maps at 10 sec (**B**) and 10,000 s (**C**) are overlaid on one monomer of the DHDDS model. The color legend shows the deuterium uptake level. Regions not covered are colored gray.

**Figure 7 biomolecules-09-00660-f007:**
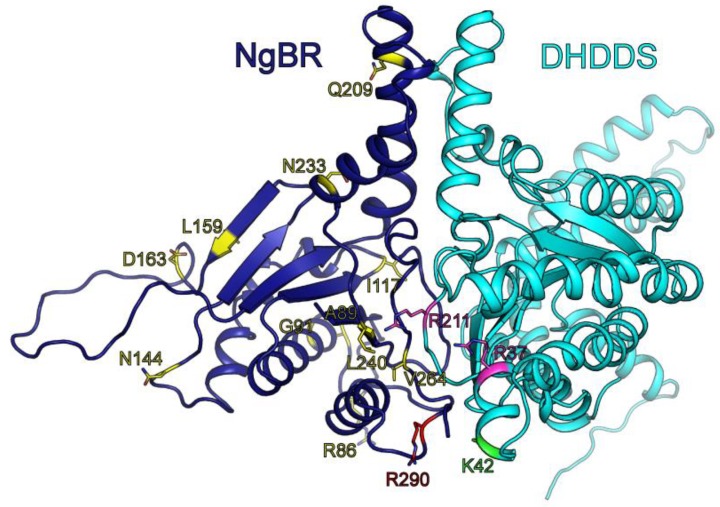
A model for the heterodimeric *cis*-prenyltransferase complex. The model of the heterodimeric complex is shown in cartoon representation. DHDDS and NgBR are colored cyan and blue, respectively. Positions with disease-associated mutations are shown as sticks and colored as follows: K42, related to retinitis pigmentosa, green; R37 and R211, related to developmental epileptic encephalopathies, magenta; R290, related to a congenital glycosylation disorder, red; R86, A89, G91, I117, N144, L159, D163, Q209, N233, L240 and V264, related to Parkinson’s disease, yellow.

**Table 1 biomolecules-09-00660-t001:** SAXS data collection and analysis.

Data Collection Parameters	
**Beamline**	ESRF BM29
**Beam geometry (mm^2^)**	0.7 × 0.7
**Wavelength (Å)**	1.0
**Q range (Å^−1^)**	0.0025–0.5
**Exposure per frame (seconds)**	1
**Temperature (°C)**	20
**Structural parameters**	
**R_g_ (Å)^1^**	31.8 ± 1.4
**D_max_ (Å)^2^**	105 ± 11
**Porod volume [from *P*(*r*)] (10^3^ Å^3^)**	132.8
**Estimated mass (kDa)^2^**	78.1 ± 7.8
**Porod exponent**	3.6
**χ^2^ model**	1.03
**Software employed**	
**Primary data reduction**	AUTOMAR
**Data processing**	PRIMUS, GNOM, ScÅtter

^1^ ± S.E. ^2^ ± 10% (estimated range).

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
