# Peer review of "Structural Characterization of Full-Length Human Dehydrodolichyl Diphosphate Synthase Using an Integrative Computational and Experimental Approach"

_biomolecules, 2019, doi:10.3390/biom9110660_

Round 1

Reviewer 1 Report

This MS describes the modeling of the unknown N-terminal and C-terminal parts of human DHDDS, which forms a homodimer with reduced activity. The authors also included some data of SAXS and HDX-MS, which however did not offer any direct evidence for the correctness of the model. Because the model is questionable, perhaps the authors may consider a change of the title to “Characterization of DHDDS by using a hybrid approach” or something like that.

About one half of the abstract was devoted to reviewing DHDDS for its biological importance. This lengthy introductory part should be boiled down to two or three short sentences. Instead, other findings in this MS should be summarized here. For example, the SEC-MALS results that showed homodimer formation, as well as the lower kcat value than the heterodimer’s, can be included.

The models in Figures 5A and 6B/C have two non-symmetrical N-terminal helices. The C-terminal parts are also a bit different from one another. Why? Does it need to be disposed like this for consistence with the experimental results? Judging by the close proximity, is there significant possibility that the N-terminus associates in a preferred way with the C-terminal helix-turn-helix (or helix-loop-helix) motif?

It would be informative to indicate the locations of secondary structural elements in Figure 6A, possibly by shading the sequence in yellow and cyan for the predicted helices and strands.

Line 168-169: The primary citations for PDB 5hxn and 6acs should be included in the reference list.

Line 257: The rationale in assigning a helix-turn-helix (or helix-loop-helix) motif to the C-terminal region is unclear. Please explain the modeling strategy in a comprehensive manner. Is this region also predicted to contain alpha helices like the N-terminus? Are there favored interactions within the HTH (or HLH) motif? (Please use only one, HTH or HLH.)

Line 269: The images presented in Figures 5 and 6 indicate that the C-terminal HTH (or HLH) motif protrudes from the dimer interface, rather than tightly packed against it. Please re-write the description, or revise the model itself.

Line 349-351: This is a wild guess, as long as the correctness of the model is questionable. Is there significant sequence homology between the dimerization region of helix 5/6 and the C-terminal region? Please elaborate or remove.

Line 370: Replace “unstructured” with “surface”.

Author Response

We would like to thank you the reviewer for the valuable suggestions, criticism and comments that helped us improve ourrevised manuscript. As specified below, we have fully addressed all the points raised by the reviewer and revised themanuscript accordingly:

This MS describes the modeling of the unknown N-terminal and C-terminal parts of human DHDDS, which forms a homodimer with reduced activity. The authors also included some data of SAXS and HDX-MS, which however did not offer any direct evidence for the correctness of the model. Because the model is questionable, perhaps the authors may consider a change of the title to “Characterization of DHDDS by using a hybrid approach” or something like that.

Response: We accept the reviewer’s suggestion and changed the title accordingly.

About one half of the abstract was devoted to reviewing DHDDS for its biological importance. This lengthy introductory part should be boiled down to two or three short sentences. Instead, other findings in this MS should be summarized here. For example, the SEC-MALS results that showed homodimer formation, as well as the lower kcat value than the heterodimer’s, can be included.

Response: Thank you for this comment. We have shortened the introductory part and included the SEC-MALS and enzymatic assay results.

The models in Figures 5A and 6B/C have two non-symmetrical N-terminal helices. The C-terminal parts are also a bit different from one another. Why? Does it need to be disposed like this for consistence with the experimental results? Judging by the close proximity, is there significant possibility that the N-terminus associates in a preferred way with the C-terminal helix-turn-helix (or helix-loop-helix) motif?

Response: Symmetry was not a restraint in our computational modelling, and it should not be enforced in our opinion since some degree of movement is expected in both termini. That being said, the deviations within our models ensemble, all consistent with the experimental data, are slight. Thus, this small degree of asymmetry is not required for consistency with the experimental results. There is no preference for close proximity between the termini in our model. They may interact, of course, but we don’t have evidence to support this hypothesis.

It would be informative to indicate the locations of secondary structural elements in Figure 6A, possibly by shading the sequence in yellow and cyan for the predicted helices and strands.

Response: We thank the reviewer for this suggestion, but think that it may be confusing together with the heat maps. The secondary structure elements are clearly indicated in Figure 1.

Line 168-169: The primary citations for PDB 5hxn and 6acs should be included in the reference list.

Response: We added the references, thank you.

Line 257: The rationale in assigning a helix-turn-helix (or helix-loop-helix) motif to the C-terminal region is unclear. Please explain the modeling strategy in a comprehensive manner. Is this region also predicted to contain alpha helices like the N-terminus? Are there favored interactions within the HTH (or HLH) motif? (Please use only one, HTH or HLH.)

Response: Helix-turn-helix motif was not assigned, this structure was obtained using RaptorXcontact, which is using co-evolutionary information to predict residue-residue contacts (http://raptorx.uchicago.edu/ContactMap/). This is now explained in the methods section. So this motif is consistent with co-evolution data that shows covariation (preferred interactions) between residues of the two helices. Moreover, secondary structure predictors predict the HTH there. We now use the term helix-turn-helix throughout the revised manuscript.

Line 269: The images presented in Figures 5 and 6 indicate that the C-terminal HTH (or HLH) motif protrudes from the dimer interface, rather than tightly packed against it. Please re-write the description, or revise the model itself.

Response: We have rephrased this sentence – our meaning was that the motif is packed against the cis-prenyltransferase domain, not the dimerization interface.

Line 349-351: This is a wild guess, as long as the correctness of the model is questionable. Is there significant sequence homology between the dimerization region of helix 5/6 and the C-terminal region? Please elaborate or remove.

Response: We removed the statement regarding helices 5 and 6. However, the position of the motif at the protein surface may have a role, and this is a hypothesis which should drive future studies.

Line 370: Replace “unstructured” with “surface”.

Response: Corrected, thank you.

Reviewer 2 Report

The manuscript by Lisnyansky Barel et al. presents a model for the homodimer of dehydrodolichyl diphosphate synthase (DHDDS) and of DHDDS with its interaction partner NgBR. First, the authors confirmed the predicted homodimeric nature of DHDDS using SEC-MALS and investigated the catalytic activity of the protein in absence of NgBR, concluding that NgBR acts as an allosteric activator to DHDDS while having no effect on the affinity towards the substrate. DHDDS was then characterized by SAXS and hydrogen/deuterium exchange coupled to mass spectrometry. A model of the DHDDS homodimer, in which the C-terminal domain was predicted to be structured, was found to be in agreement with SAXS data. HDX-MS data confirmed the homodimerization interface and the structured state of the C-terminal domain. In addition to the DHDDS dimer, a model for the DHDDS-NgBR heterodimer was constructed based on the homodimer model and a homology model for NgBR generated from the yeast ortholog.

In summary, the data presented in the manuscript provide new insight into the structure of DHDDS, a protein that has not been extensively studied from a structural point of view, but that has important biological functions due to its role in N-glycosylation and its connection to several diseases such as retinitis pigmentosa and epileptic encephalopathies.

I have only a few minor comments that should be corrected before publication.

Page 4, line 156: Quadrupole-time-of-flight (not fly)

Page 4, line 160: If no enzyme specificity is defined, how can a missed cleavage be considered for HDX-MS data analysis?

Page 5, lines 187 and 199: monodisperse, not monodispersed

Page 5, lines 192 and 194: dipole, not diploe. Specifically, the description of the MALS principle (probably "inspired" by the explanations on the Wyatt website, https://www.wyatt.com/library/theory/understanding-multi-angle-static-light-scattering.html) needs some rephrasing. The induced dipole is not "re-radiated", it re-radiates light, and the scattered light (not the dipole as written in the manuscript) is measured.

Pages 6 and 7: The values given for the radius of gyration are not consistent between the text (line 232) and Table 1.

Page 8, line 279: The term "peptic peptides" may not be clear to readers that are unfamiliar with HDX-MS. The information that pepsin is used as the protease could be added to the paragraph above, line 274.

Page 8, line 283: The aggregation of the protein upon full deuteration is not really "discussed" in the method section, it is simply stated there that the protein aggregates. Is this a consequence of the long incubation times?

Author Response

We would like to thank you the reviewer for the valuable suggestions, criticism and comments that helped us improve ourrevised manuscript. As specified below, we have fully addressed all the points raised by the reviewer and revised themanuscript accordingly:

The manuscript by Lisnyansky Barel et al. presents a model for the homodimer of dehydrodolichyl diphosphate synthase (DHDDS) and of DHDDS with its interaction partner NgBR. First, the authors confirmed the predicted homodimeric nature of DHDDS using SEC-MALS and investigated the catalytic activity of the protein in absence of NgBR, concluding that NgBR acts as an allosteric activator to DHDDS while having no effect on the affinity towards the substrate. DHDDS was then characterized by SAXS and hydrogen/deuterium exchange coupled to mass spectrometry. A model of the DHDDS homodimer, in which the C-terminal domain was predicted to be structured, was found to be in agreement with SAXS data. HDX-MS data confirmed the homodimerization interface and the structured state of the C-terminal domain. In addition to the DHDDS dimer, a model for the DHDDS-NgBR heterodimer was constructed based on the homodimer model and a homology model for NgBR generated from the yeast ortholog.

In summary, the data presented in the manuscript provide new insight into the structure of DHDDS, a protein that has not been extensively studied from a structural point of view, but that has important biological functions due to its role in N-glycosylation and its connection to several diseases such as retinitis pigmentosa and epileptic encephalopathies.

I have only a few minor comments that should be corrected before publication.

Page 4, line 156: Quadrupole-time-of-flight (not fly)

Response: Corrected, thank you.

Page 4, line 160: If no enzyme specificity is defined, how can a missed cleavage be considered for HDX-MS data analysis?

Response: When using non-specific digestion, the missed cleavage parameter is disregarded by the software, but is still given as input. The default value is 1. We removed the missed cleavage parameter from the methods section for clarity.

Page 5, lines 187 and 199: monodisperse, not monodispersed

Response: Corrected, thank you.

Page 5, lines 192 and 194: dipole, not diploe. Specifically, the description of the MALS principle (probably "inspired" by the explanations on the Wyatt website, https://www.wyatt.com/library/theory/understanding-multi-angle-static-light-scattering.html) needs some rephrasing. The induced dipole is not "re-radiated", it re-radiates light, and the scattered light (not the dipole as written in the manuscript) is measured.

Response: We have rephrased the explanation. Thank you.

Pages 6 and 7: The values given for the radius of gyration are not consistent between the text (line 232) and Table 1.

Response: Thank you for the comment. The correct value was the one mentioned in the text and we corrected the table accordingly.

Page 8, line 279: The term "peptic peptides" may not be clear to readers that are unfamiliar with HDX-MS. The information that pepsin is used as the protease could be added to the paragraph above, line 274.

Response: We have rephrased line 274 according to your suggestion: “followed by proteolytic digestion with pepsin and separation of the peptic peptides using reverse-phase chromatography”

Page 8, line 283: The aggregation of the protein upon full deuteration is not really "discussed" in the method section, it is simply stated there that the protein aggregates. Is this a consequence of the long incubation times?

Response: Yes, it is due to the long incubation, as now explained in the methods section.